# Protective Effect of Dexmedetomidine against Hyperoxia-Damaged Cerebellar Neurodevelopment in the Juvenile Rat

**DOI:** 10.3390/antiox12040980

**Published:** 2023-04-21

**Authors:** Robert Puls, Clarissa von Haefen, Christoph Bührer, Stefanie Endesfelder

**Affiliations:** 1Department of Neonatology, Charité—Universitätsmedizin Berlin, Augustenburger Platz 1, 13353 Berlin, Germany; robert.puls@charite.de (R.P.); christoph.buehrer@charite.de (C.B.); 2Department of Anesthesiology and Intensive Care Medicine, Charité—Universitätsmedizin Berlin, 13353 Berlin, Germany; clarissa.von-haefen@charite.de

**Keywords:** hyperoxia, oxidative stress, cerebellum, dexmedetomidine, postnatal developing brain, newborn rat

## Abstract

Impaired cerebellar development of premature infants and the associated impairment of cerebellar functions in cognitive development could be crucial factors for neurodevelopmental disorders. Anesthetic- and hyperoxia-induced neurotoxicity of the immature brain can lead to learning and behavioral disorders. Dexmedetomidine (DEX), which is associated with neuroprotective properties, is increasingly being studied for off-label use in the NICU. For this purpose, six-day-old Wistar rats (P6) were exposed to hyperoxia (80% O_2_) or normoxia (21% O_2_) for 24 h after DEX (5 µg/kg, i.p.) or vehicle (0.9% NaCl) application. An initial detection in the immature rat cerebellum was performed after the termination of hyperoxia at P7 and then after recovery in room air at P9, P11, and P14. Hyperoxia reduced the proportion of Calb1+-Purkinje cells and affected the dendrite length at P7 and/or P9/P11. Proliferating Pax6+-granule progenitors remained reduced after hyperoxia and until P14. The expression of neurotrophins and neuronal transcription factors/markers of proliferation, migration, and survival were also reduced by oxidative stress in different manners. DEX demonstrated protective effects on hyperoxia-injured Purkinje cells, and DEX without hyperoxia modulated neuronal transcription in the short term without any effects at the cellular level. DEX protects hyperoxia-damaged Purkinje cells and appears to differentially affect cerebellar granular cell neurogenesis following oxidative stress.

## 1. Introduction

Prematurity is associated with lifelong impairments in learning, language, cognition, and socio-emotional behavioral disorders, such as attention deficit hyperactivity disorder (ADHD) and autism spectrum disorders (ASDs) [1,2,3,4,5,6,7]. Functional and structural impairments arise from the immature, developing brain being challenged too early [8,9].

The vulnerability of the developing brain and neuromorbidity stem from both extrinsic and intrinsic factors. Extrinsic vulnerability relates to the harmful effects of perinatal morbidities on the brain, such as drugs and anesthetics; however, oxidative stress that is triggered by the oxygen transition during birth from the intrauterine hypoxia to the extra-uterine hyperoxia as well as additional oxygen substitution in respiratory instabilities of the preterm infant are also detrimental. The intrinsic factors are due to the structural and molecular immaturity of the developing brain, which depends on the gestational age [10].

The cerebellum assumes a critical role in motor and cognitive functions [11]. Several motor and cognitive deficits that may persist over the long term [12,13] have been associated with decreased cerebellar size of preterm infants. The cerebellum exhibits the greatest growth and development during the last trimester of pregnancy, and is thus highly vulnerable to neuromorbidity [14,15].

The development of motor disorders and cognitive difficulty are sufficiently associated with the disturbance of postnatal cerebellar development. This is triggered not only by premature birth per se, which leads to focal lesions of the cerebellum, but also by influences that disrupt the expansion and development of the well-orchestrated regulation of neuronal networks throughout the cerebellum [16]. During postnatal cerebellar development, neural progenitor cells undergo essential processes of mitosis, cell fate specification, migration, neurite outgrowth, and spin formation [17]. It is a chronological period of high neuroplasticity in which cellular populations and neuronal networks are extremely sensitive to environmental influences and compensation can be rather subtle and long-lasting. The main cell populations involved in cerebellar neurogenesis are Purkinje cells (PC) and granular cells (GC) [18,19]. GCs are characterized by a high proliferative capacity in the external granular layer (EGL) before radially migrating to the internal granular layer (IGL) and maturing. PCs guide the development of rudimentary circuits. The signaling mechanisms between GCs and PCs control both GC development and the circuitry of neuronal signals of PCs [20,21,22]. The vulnerable phase of granule cell neurogenesis occurs postnatally in humans between 20 and 40 weeks after conception and thus primarily affects all preterm infants, but because of immaturity, injury severity usually correlates with extreme and very low birth weight. In rodents, granule cell neurogenesis peaks between the first and second weeks of life [17,23,24,25]. A detailed description of cerebellar neurogenesis can be found in the review article by Komuro et al. [26], and Figure 1 presents a simplified illustration.

The unintended oxygen over-supply of preterm infants by birth per se and possible oxygen addition during respiratory instabilities induce oxidative stress, the oxygen-toxic effects of which, in the presence of immature antioxidant enzyme production, may cause disease in the preterm infant [29,30]. Neurodevelopmental deficits and injuries of the immature cerebellum can be multicausal [31]. Oxidative stress has been shown to be a critical factor. Optimal oxygen concentrations in the care of premature infants have not yet been conclusively standardized [32,33]. Hyperoxia, hypoxic ischemia, infections, and inflammatory processes of different genesis cause oxidative stress [34,35]. The medically necessary sedation of extremely preterm infants leads to exposure to anesthetics, analgesics, and sedatives both during procedures and in the neonatal intensive care unit. Consequently, all of these factors, including prematurity and oxidative stress, correlate with a higher risk of neurodevelopmental disorders [36].

Dexmedetomidine is a potent and highly selective α2-adrenergic receptor agonist that has sedative, analgesic, and antianxiety effects; however, it does not suppress ventilation [37], and it reduces the need for additional sedation and has demonstrated opioid- and benzodiazepine-reducing effects under sedation [38,39,40]. One important property of DEX is its potential neuroprotection on the developing brain, which has been the subject of diverse experimental work in preclinical models. Neonatal animal models of hypoxia/ischemia have demonstrated that dexmedetomidine acts as a potent neuroprotector via stimulation of the α2-adrenergic receptor [41] and reduces lesions in cortical and white matter, as well as being associated with improved neurological function [42]. Other mechanisms of neuroprotection by dexmedetomidine, as demonstrated by previous work in the newborn rats, include reductions in oxidative stress and thus, revealing anti-inflammatory and anti-apoptotic effects [43,44].

To date, little is known about the effects of dexmedetomidine in combination with oxidative stress on the developing brain. In this study, the effect of dexmedetomidine on postnatal cerebellar neurogenesis after acute injury and subsequent recovery will be investigated in a hyperoxia-mediated injury model of a neonatal rat.

## 2. Materials and Methods

### 2.1. Animal Welfare

Wistar rats were term-paired (Janvier Labs, Le Genest-Saint-Isle, France) and kept in single cages under standardized conditions (constant 12 h/12 h light-dark cycle, room temperature, 60% relative humidity, and ad libitum to food and water). In all of our animal experiments [28,43,44,45], newborn pups were kept with a lactating mother for the duration of the experiment. The animal procedures were assessed and approved by the local animal welfare authorities (LAGeSo, approval number G-0145/13 and G-139/18) and complied with institutional guidelines and ARRIVE guidelines.

### 2.2. Oxygen Exposure and Drug Administration

As previously described [43,44], the rat pups were randomized according to litter and sex. This was conducted four days after birth. Dams with pups were then exposed to either hyperoxia (80% O_2_, OxyCycler BioSpherix, Lacona, NY, USA) or normoxia (21% O_2_), starting from postnatal day (P)6 to P7. Application of body weight-adapted dexmedetomidine (5 µg/kg; DEX; dexdor^®^, Orion Pharma, Espoo, Finland; dissolved in phosphate-buffered saline (PBS)) was administered intraperitoneally (i.p.; 100 µL/10 g body weight). Drugs and vehicle (0.9% saline) were administered once 15 min before the start of the oxygen exposure. Rats were divided into a control group (NO: 21% O_2_, normoxia) and three experimental groups (NOD: 21% O_2_ with 5 µg/kg DEX; HY: 80% O_2_ (OxyCycler BioSpherix, Lacona, NY, USA) with 0.9% saline; HYD: 80% O_2_ with 5 µg/kg DEX), each with 6 pups. The analysis times were set after the end of oxygen exposure (P7) and further after recovery in room air at P9, P11, and P14. The animal experiments were carried out as previously described [43,44].

### 2.3. Tissue Preparation

Using an i.p. injection of ketamine (100 mg/kg), xylazine (20 mg/kg), and acepromazine (3 mg/kg), the pups were anaesthetized at P7, P9, P11, and P14 and subsequently transcardially perfused, as previously described [28,43,44,45]. The entire cerebellar tissue was removed immediately after decapitation and opening of the skull. Cryopreservation in liquid nitrogen and storage at −80 °C was performed for the tissues after perfusion with PBS (pH7.4) preparing for gene expression studies. Perfusion with PBS (pH 7.4) followed by 4% paraformaldehyde (PFA, pH7.4) was to be performed for the subsequent immunohistochemical analyses. Post-fixation with PFA at 4 °C for 24 h prepared the samples for dehydration with ascending ethanol series and embedding in paraffin.

### 2.4. RNA Extraction and Quantitative Real-Time PCR

As previously described for the extraction of RNA [28,46], the total RNA from whole snap-frozen cerebellum of each animal was homogenized in peqGOLD RNAPure™ (PEQLAB Biotechnologie, Erlangen, Germany), according to the manufacturer’s instructions. After DNase treatment, 2 µg of RNA was reverse transcribed.

Quantitative PCR was performed using the qPCR BIO Mix Hi-ROX (NIPPON Genetics Europe, Düren, Germany). Gene expression analyses for selected target genes (Table 1) were performed using the StepOnePlus real-time PCR system (Applied Biosystems, Carlsbad, CA, USA) analyzed according to the 2^−ΔΔCT^ method [47], and with a constant amplification program (50 °C for 2 min, 94 °C for 2 min, 40 cycles at 94 °C for 5 s, and 62 °C for 25 s). In a 11 µL reaction mixture consisting of 5 μL qPCR master mix, 2.5 μL 1.25 μM of each oligonucleotide primer, 0.5 μL 5 μM of probe, and 3 μL of cDNA template (17 ng), amplification was performed using fluorescent reporters and quencher-labeled probes (BioTez Berlin Buch GmbH, Berlin, Germany). The expression of target genes was analyzed using hypoxanthine-guanine phosphoribosyl-transferase (HPRT) as an internal reference.

### 2.5. Immunohistochemistry

Paraffin-embedded whole cerebellums were cut into 5 µm-thick serial slices and subsequently mounted on SuperFrost Plus-coated slides (Menzel, Braunschweig, Germany) with three consecutive slices each, as previously described [28,46]. After deparaffinization in Roti-Histol (Carl Roth, Karlsruhe, Germany), twice for 10 min each and rehydration in descending ethanol concentration for 3 min each, heat-induced antigen retrieval was applied to the paraffin sections in citrate buffer (pH 6.0) in a microwave oven for 10 min at 600 W. To block the non-specific binding sites, the sections were incubated with appropriate blocking solutions for 1 h at room temperature (calbindin/DAPI: 3% bovine serum albumin (BSA), 0.05% TW-20 and 0.1% Triton X-100 in PBS; Pax6/PCNA/DAPI: 3% BSA and 0.2% Triton X-100 in PBS). After blocking, the sections were incubated overnight with specific primary antibodies at 4 °C after washing the sections once in PBS (monoclonal mouse anti-calbindin (1:500, Abcam, ab75524) or polyclonal rabbit anti-Pax6 (1:200, LifeSpan Bioscience, LS-C179903) with monoclonal mouse anti-PCNA (1:500, Abcam, ab29) diluted in antibody diluent (Zymed Laboratories, San Francisco, CA, USA)). Then, the sections were washed with PBS 3 times, with each rinse performed for 5 min. Next, slices were incubated for 1 h at room temperature with secondary Alexa Fluor 488-conjugated goat anti-mouse IgG (Thermo Fisher Scientific, Dreieich, Germany) or Alexa Fluor 594-conjugated goat anti-rabbit IgG (Thermo Fisher Scientific), with consistent 1:200 dilution in antibody diluent (Zymed Laboratories). Washing once in PBS was followed by incubation at room temperature for 10 min with 4′,6-diamidino-2-phenylindole (DAPI, Sigma, diluted 1:2000 in PBS) for nucleus staining. After washing three times with PBS, the tissue sections were cover slipped (Shandon Immu-Mount, Thermo Fisher Scientific).

For the analyses and quantification of middle cerebellar sections, the digitalization was performed using a Keyence BZ 9000 compact fluorescence microscope with BZ-II Viewer software and BZ-II Analyzer software (Keyence, Osaka, Japan). The images were analyzed blindly with 20x objectives after the individual files were automatically merged for each RGB color. For each time point, the images for all experimental groups were generated at the same time, considering the same exposure time and the same contrast/brightness parameters. As described previously [28], the analyses for each animal were performed using four non-overlapping separate images of posterior lobules IV/V, VI, and/or VII, including two external cortices and two inner loops of the cerebellar cortex. For quantification of Purkinje cells (calbindin+), granule neurons (Pax6+), and proliferation marker (PCNA), four 100 µm regions of lobules were quantified for each section and counted manually using Adobe Photoshop software 22.0.0 (Adobe Systems Software Ireland Limited, Dublin, Republic of Ireland) with minimal previous manipulation of contrast. Identical digital images of the calbindin-counting images were used to determine the depth of the molecular layer. For determining the depth of the molecular layer, the Calb1-postive stained Purkinje cells were used. The dendrite length of four randomly selected Purkinje cells served as the criterion by measuring the primary dendrite from the cell soma to the surface of the molecular layer. We calculated the mean of all sections of the same animal. Each individual animal was calculated in relation to the mean of the control group and then plotted as the mean of each experimental group compared to the control group (100% mean, raw data are highlighted in the legends). The assigned numbers of Calb1 cell counts of ROI are listed in Appendix A.

### 2.6. Statistical Analyses

All statistical analyses were conducted using the GraphPad Prism 8.0 software (GraphPad Software, La Jolla, CA, USA). As previously described [28], data were analyzed using a multivariate repeated measures analysis of variance (ANOVA). Model selection between different variance/covariance structures was based on a partially non-Gaussian distribution with the Kruskal-Wallis test or assuming that groups do not have equal variances with the Brown-Forsythe test. Depending on which ANOVA test was used, multiple comparisons of means were carried out using Bonferronis, Dunn’s, or Dunnett’s T3 post hoc test. All statistics were evaluated with a *p*-value of <0.05. GraphPad Prism Software was used for the generation of graphs. Data are presented as box and whisker plots, with the line representing the median while whiskers show the data variability outside the upper and lower quartiles.

## 3. Results

### 3.1. Dexmedetomidine Protects Hyperoxia-Induced Impairment of Purkinje Cells

Calbindin1 (Calb1) is a specific marker for Purkinje cells of the cerebellar cortex [17]. To evaluate an impact of acute hyperoxia in the dynamic phase of cerebellar development, the changes in the amount of Calb1-positive Purkinje cells and the length of their dendrites in the molecular layer (ML) were analyzed after hyperoxia, with and without DEX. What could be perceived subjectively from the immunohistochemical staining (Figure 2), namely that, especially directly after hyperoxia and also after recovery to room air, the PCs seemed to be impaired, was confirmed in the quantitative analysis (Figure 3). Hyperoxia resulted in a significant reduction in the number of Calb1-positive PCs at P7 and P9 (Figure 3A) and in dendrite length at P7 and P14 (Figure 3B) compared with normoxic control animals. Both parameters normalized up to P14. Pretreatment with DEX was able to constrain the hyperoxia-induced impairments for both PCs and dendrite growth. DEX under normoxic exposure impaired dendrite growth at P7, although this effect attenuated by day P14. The complete data of the values related to the 100% values of the control groups as well as probably the Calb1 cell counts per ROI are shown in Appendix A.

### 3.2. Dexmedetomidine Reduces the Prolonged Damage to Mitotic Granular Precursor Cells Caused by Hyperoxia

An important cerebellar developmental stage is the rapid proliferation of granule cell precursors. We firstly investigated the proportion of Pax6-expressing cells and the cerebellar proliferation capacity using the proportion of Pax6-positive cells and by means of the proliferation marker PCNA, and secondly, we examined the proportion of GPCs with regard to hyperoxia damage by proliferating Pax6 cells. We observed a marked reduction in Pax6- and PCNA-positive immunocytochemistry at P7, immediately after hyperoxic insult, compared with the control cerebellum, whereas DEX impressively abolished this downregulation (Figure 4). Indeed, Pax6 expression is dysregulated in the EGL at P7 of the developing cerebellum after high levels of oxygen (Figure 5A). This is consistent with a reduced proliferative capacity of the cerebellum at P7 and P11 (Figure 5B), although this effect was prevented by DEX only at P7 (Figure 5A,B). However, when the proportion of proliferating Pax6-positive GPCs is considered in comparison (Figure 5C), 24 h of hyperoxia resulted in a drastic reduction in dividing granular progenitor cells that persisted until P14 despite recovery. A single application of DEX completely prevented this sustained damage to GPC at the cellular level (Figure 5C). However, the effect of this single application of DEX without oxidative stress was also strikingly apparent, as the proportion of mitotic GPCs was equally sustainably reduced on the postnatal days 9 and 14, i.e., at least 2 days after potential damage (Figure 5C). The complete data are presented in Appendix A.

### 3.3. Dexmedetomidine Differentially Modulates Neuronal Transcription Factors of Purkinje Cells, Neurotrophins, Granular Precursor Cells, and Granular Cells

The molecular layer provides a compartment for the synaptic connections between the dendritic tree of the Purkinje cells and the parallel fibers of the granule cells. The Purkinje cells are connected to the white matter and connect to the cerebellar nuclei. The activity of the granule cells is modulated, among others, by the unipolar brush cells. A complex neuronal network connects the cerebellar neurons and enables their tasks. The connectivity and formation in understanding cerebellar neurogenesis, migration, and dendritogenesis is orchestrated by a manifold transcriptome [48].

#### 3.3.1. Hyperoxia-Induced Damage on Transcriptional Purkinje Cell-Associated Factors and Neurotrophins Does Not Consistently Reflect Cellular Protection with Dexmedetomidine

Gene expression of *Calb1* and *Shh* was significantly reduced after acute hyperoxia exposure at P7 and P9 (Figure 6A), with *Shh* remaining sustainably reduced until P14 (Figure 6B). *Calb1* was overexpressed twofold after hyperoxia at P11 (Figure 6A). DEX had no consistent protective effect to attenuate hyperoxia-mediated damage. Gene regulation occurred for *Calb1* at P7 and P11 as well as P14 (Figure 6A). The normalization of *Shh* by pretreatment with DEX succeeded only at P11 (Figure 6B). Animals who were not exposed to hyperoxia after DEX administration showed a reduced expression for *Calb1* at P9 (Figure 6A), and for *Shh* at P7 and P14 (Figure 6B).

Neurotrophins regulate the survival and differentiation of different populations of cerebellar neurons during development. The gene expression of *BDNF* (Figure 6C) and *NT3* (Figure 6E) significantly decreased after acute hyperoxia exposure at P7 and/or P14. DEX showed an expression-increasing effect for *BDNF* (Figure 6C) at P14 after reduction, as well as for *NGF* at P7 and P14 (Figure 6D). Hyperoxia reduced *NT3* expression at P7, but increased it at P11 and P14 (Figure 6E). DEX had no effect on hyperoxia-induced neurotrophin expression, except at P14 (Figure 6C–E). DEX under ambient conditions showed identical expression patterns as under hyperoxic insult (Figure 6C–E). The complete data are presented in Appendix A.

#### 3.3.2. Hyperoxia Decreases Proliferation and Migration Mediators of Granular Precursor Cells and Dexmedetomidine Only Improves Proliferation Impairment

The gene expression of proliferation-associated factors of GPCs, *CycD2*, and *Pax6* significantly decreased after acute hyperoxia exposure at P7 and/or persisting to P9 (Figure 7A,B). *CycD2* was overexpressed at P11 and P14 after recovery in room air. DEX was able to counteract the hyperoxia insult for *CycD2* and *Pax6* at P7, respectively, or reduce the overexpression of *CycD2* at P11 and P14 to normoxia levels (Figure 7A). Similarly, DEX reduced *Pax6* expression at P14 after hyperoxia (Figure 7B). DEX without hyperoxia insult reduced *CycD2* at P7 and P9 and *Pax6* at P7, while DEX also induced *CycD2* and *Pax6* expression at P11 and/or P14 (Figure 7A,B).

One-day hyperoxia decreased the migration-associated factors of GPCs, *Lmx1α*, and *Sema6a* at P7 and additionally for *Lmx1α* at P9 (Figure 7C,D). Overexpression occurred after hyperoxia and room air at P11 for *Sema6a* (Figure 7D). DEX alone had a reducing effect on *Lmx1α* and *Sema6a* at P7 under control conditions and did not have a protective effect per se against hyperoxia-induced downregulation (Figure 7C,D). The complete data are presented in Appendix A.

#### 3.3.3. Dexmedetomidine Has Only Low Effects on Hyperoxia-Injured Mediators of Cerebellar Development and the Survival of Granular Cells

High oxygen exposure significantly reduced the transcription of *NeuroD1* and *NeuN* at P7 and/or P9 (Figure 8A,B). Induced expression after hyperoxia occurred for *NeuroD1*, *NeuN*, and *Chd7* at P11 (Figure 8A,B,D). DEX decreased the expression of *NeuroD1* at P11, as well as of *NeuN* and *Chd7* at P14 under hyperoxia, but not at the acute hyperoxia termination time at P7 (Figure 8A,B,D). Increased expression was achieved under hyperoxia with DEX for *Prox1* at P9 (Figure 8C) and for *Chd7* at P7 (Figure 8D). DEX under normoxia reduced the expression of *NeuroD1* and *NeuN* at P7 and/or P9 (Figure 8A,B) and increased *NeuroD1* and *Prox1* at P11 (Figure 8C,D). The complete data are presented in Appendix A.

#### 3.3.4. Hyperoxia-Damaged Neuronal Transcripts of Granular Cells and Dexmedetomidine Protected Only the Differentiation-Dependent Factor NeuroD2

*Pax2* and *Sox2* mRNA transcription is reduced under hyperoxic conditions after acute hyperoxia at P7 (Figure 9A,D). Hyperoxia reduced *Syp* expression at P7 and sustained expression at P9 and P11 (Figure 9C). *NeuroD2* was reduced at P7 and P9, and induced at P11 by a one-day exposure to high oxygen (Figure 9D). DEX as a pretreatment of hyperoxic insult did not affect the expression of *Pax2*, *Syp*, and *Sox2* (Figure 9A,C,D), whereas *NeuroD2* was significantly affected by DEX at P7, P11, and P14 (Figure 9B). Notably, DEX under normoxia had a reducing effect on the transcription of all factors of granular cell differentiation and survival (*Pax2*, *NeuroD2*, *Syp*, and *Sox2*), primarily at P7 (Figure 9). DEX had the most sustained effect on *Syp* at P7, persisting until P14 (Figure 9C). The complete data are presented in Appendix A.

#### 3.3.5. Dexmedetomidine Counteracts the Hyperoxia-Inhibitory Effect for Unipolar Brush Cell- and Cerebellar Nuclei Neuron-Associated Transcripts

Hyperoxia reduced both *Tbr1* and *Tbr2* mRNA expression at P7, and at P9 or P14 (Figure 10). Pretreatment with DEX before hyperoxia exposure resulted in a dramatic increase in expression, twofold above the control level, for *Tbr1* and *Tbr2* after acute hyperoxia as well as after recovery at room air to P14 (Figure 10). DEX under normoxia reduced *Tbr1* at P7 (Figure 10A) and increased both *Tbr1* and *Tbr2* at P11 and P14 (Figure 10A,B). The complete data are presented in Appendix A.

## 4. Discussion

In the present study, we provided evidence for the impact of preventive DEX application prior to hyperoxic damage to the postnatal rat cerebellum, as toxically high oxygen concentration resulted in a loss of PCs, dendritic PC abnormalities, and delayed maturation of GCs. DEX was able to reduce the cellular sequelae of oxidative stress, with higher benefits to cerebellar transcripts of GPC and GC.

The preclinical findings in newborn rodents are in agreement with clinical data from premature infants, and they clearly show that early exposure to anesthetic, analgesic, and sedative agents causes neuroapoptosis and can impair long-term neurodevelopment [36,49,50,51]. What is decisive for this sensitivity to sedatives, such as opioids and benzodiazepines, is certainly the high vulnerability of the immature brain in a phase of development characterized by neuronal maturation and differentiation, migration, and synaptogenesis during a phase of rapid brain growth [52]. The situation of the premature infant with regard to neurodevelopment is more vulnerable even without medically necessary interventions, since oxidative stress [29,53,54,55], with the underdeveloped antioxidative defense system of premature infants [34], can per se impair the neurological outcome. The combination of oxidative stress and sedatives with medical indications on the developing brain is a challenge.

Preterm birth and the resulting morbidities and side effects of medical treatments are major causes of neurodevelopmental disorders. Processes of complex brain development are being considered in more detail, and there is growing evidence that the cerebellum, a brain region involved in motor coordination, perception, learning, memory, and social communication [14,15], which is impaired by premature birth, is crucial for the adverse neurodevelopment of preterm infants [56,57]. Based on the high vulnerability of the cerebellum, which undergoes a series of dynamic and anatomical developmental changes especially in the third trimester of pregnancy or in the first two postnatal weeks of rodents, there is increasing evidence of cognitive, social, and motor deficits when cerebellar development is disturbed by non-physiological events [15,17,23,24,25,58].

As demonstrated in this study, the proportion of Purkinje cells and dendrite sprouting was reduced by oxidative stress in the developing cerebellum. Only a short exposure to high levels of oxygen was enough to present this damage up to 48 h later. This could be confirmed in a similar model of a 6-day-old rat with reduced dendrite formation [59]. Using a near-birth hyperoxia damage model, it was demonstrated that prolonged, high oxygen exposure led to lasting damage to the PCs [28]. This study was able to show that one-day hyperoxia caused severe and persistent degradation of mitotic GPCs. Oxidative stress reduced the proliferative capacity of the cerebellum, but also damaged the proportion of Pax6-positive progenitor cells. Shh plays an important role in GCP proliferation [60], and its role is directly linked to the cell cycle. The induction of Shh occurs during development through the expression of D-cyclins [61,62]. Both Shh and CycD2 transcripts, as well as the proliferation marker PCNA, were down-regulated by hyperoxia. This effect lasted from P11 to P14. Scheuer et al. [59] showed that tissue Shh was down-regulated to P30 in a similar rat damage model. This was accompanied by a reduced Pax6 transcript expression [59]. GC precursor cells exhibit high division rates, with a maximum at P5 [63] before the GPCs complete the amplification and migrate from the EGL to the IGL. In this phase, Shh is important for controlling the next steps of migration and differentiation. The inhibition of the Shh signaling pathway leads to a reduction in cell proliferation in the EGL and an inhibition of further cell divisions [64]. The amount of PCs is closely connected to the amount of GPCs. It has been proven that a reduced number of GC causes abnormal PC cell function, as these have excitatory granule cell synapses. However, the connections have not yet been conclusively clarified [65]. As was also proven in other experimental models, postnatal hemorrhage and hypoxia in mice resulted in decreased numbers of granule cells and consequently, abnormal motor control [57,66,67].

The proliferative and premigratory zones can be clearly assigned to the EGL during postnatal cerebellar development, which is transformed into the IGL by migration of the maturing GCs through the premigratory zone to P15 in the rodent [19,68]. The correct classification of the cells is important for the cerebellar network and cytoarchitecture of the cerebellum. Despite the sustained regulation of the Shh transcript, cellularly, there was a normalization of PCs and proliferation capacity (PCNA+ cells) per se, although mitotic progenitor cells (Pax6+/PCNA+) appeared persistently impaired. Supporting this, the phenotype of Pax6−/− mice is associated with deficits in neurite outgrowth and cell migration and thickening of the EGL. In contrast, heterozygous Pax6 mice appear to be without deficits [69,70], raising the question of the dosage molecule effect of Pax6 in a highly complex transcriptional regulatory system [71]. The Shh signaling pathway is essential, with studies in naked-ataxia mice suggesting that a broader molecular pathway and additional mechanisms regulating granule cell development during clonal expansion are involved in cerebellar neurogenesis. Thus, a general downregulation of the protein synthesis machinery could contribute to the reduced number of granule cells [72].

In this complicated process of cerebellar development and maturation of cerebellar neurons, it has been concluded that when GPC proliferation is disrupted, GC migration is often impaired [73,74]. A dysfunction of Sema6a, a factor that mediates the tangential to radial migration of postmitotic GCs along the Bergman glia, results in the reduced accumulation of mature GCs in the IGL [75]. Cerebellar formation and postnatal development are dependent on the migration-regulating factor Lmx1α [76]. Both migration-dependent factors displayed a decreased expression at the transcriptional level immediately after the end of hyperoxia or until 48 h after the end of exposure. This reduction was not sustained after recovery to room air, but it is consistent with the reduced proliferation of GC progenitor cells. Furthermore, Pax6 appears to be critical for granule cell development in general, including proliferation and differentiation, adhesion, and signaling events that are critical for Purkinje cell migration, as well as for normal granule cell migration. The early and ubiquitous expression of Pax6 in the development of granule cells appears to play a critical role in the downstream effects of Pax6 on the cerebellum. Pax6-null mice demonstrated significant effects on granule cell development as well as migration [70]. In our study, Pax6 was reduced and expressed at P7 after hyperoxic insult, which was comparable to the migration-associated factors Lmx1α and Sema6a.

Neurotrophins are important in the development of the nervous system, cerebrum, and cerebellum [77,78], and they are key players in neuroplasticity [79]. Multiple stressors, such as oxidative stress or drugs, are associated with changes in neurotrophin levels in the developing brain [28,80,81,82]. The neurotrophic factors BDNF and NT3 exhibited a decrease immediately after hyperoxia at P7, and BDNF demonstrated reduced levels after a week of recovery in room air. Motor and cerebellar phenotypes with behavioral deficits were revealed by BDNF- and NT3-deficient mice [83,84,85]. A few studies have shown that BDNF and NT3 exert a causal factor on the maturation of the cerebellar cortex, especially granule cells [86,87]. Additionally, BDNF stimulation of GPC migration can be inferred based on a BDNF concentration gradient from EGL to GCL [87]. Mutants lacking BDNF or NT3 showed defects in cerebellar growth and foliation [88]. The three main neurotrophin tyrosine kinases receptors (Trk), with the subtypes TrkA, TrkB, and TrkC for binding NGF, BDNF, and NT3, respectively. TrkB and TrkC are especially expressed on both postsynaptic PCs and presynaptic GCs, leading to paracrine-autocrine signaling in the cerebellum [89].

GPCs already begin to express NeuroD1 during differentiation, which initiates morphological GC adaptations [90]. Differentiating granule cells form axons that interact with the dendrites of Purkinje cells, while the differentiating GCs migrate from the EGL to the IGL [91]. With a NeuroD1 knockout mouse model of NeuroD1 depletion in cerebellar granule cell precursors, it was demonstrated that NeuroD1 can influence a rapid transition from proliferative precursors to granule cells [92]. Similarly, in the global absence of NeuroD1, preferential posterior cerebellar defects in granule cells have been reported [93], as well as the elimination of granule cells in the central lobes and abnormalities in Purkinje cells in a GPC-selective NeuroD1 knockout model [92]. These studies with transgene mice support the importance of interaction between Purkinje and granule cells during cerebellar development, despite the variability in phenotypes when NeuroD1 is impaired. NeuN is highly expressed, like NeuroD1, in postmitotic neurons, and is associated with differentiating GCPs [94]. NeuroD2 was also significantly affected by hyperoxia in the cerebellum, which is consistent with NeuroD1. NeuroD1 seems to be important for blocking the proliferation of mature GCPs after they are integrated into the IGL, but NeuroD2 seems to be important for their survival [90,95]. As exhibited in NeuroD2-null mutant mice, both GC survival and PC formation with dendrite sprouting were impaired [90]. Several neurotrophic factors are also involved in this process. The positive effect of BDNF on the induction of postmitotic-expressed NeuroD was demonstrated in embryonic stem cells [96].

Chd7, which is necessary for the transcription of GC-associated neuronal genes, was not modulated by acute hyperoxia, but it was increased at P11. The inactivation of Chd7 in cerebellar granule cell precursors led to cerebellar hypoplasia in mice due to impaired granule cell differentiation and abnormal Purkinje cell localization. Chd7 appears to be important for granule neuron differentiation due to its function in maintaining open chromatin structure and recruiting DNA topoisomerase, which is required for neuronal gene activation. An epigenetic mechanism facilitated by the Chd7-mediated regulation of gene transcription is proposed by Feng and colleagues [97]. It is also supported by the fact that Chd7 is strongly expressed in GCs during cerebellar development and is maintained in the mature cerebellum. However, PCs do not express Chd7 [97,98]. In a conditional knockout of Chd7 in granule cell precursors of the mouse cerebellum, Chd7 depletion appears to affect perinatal cerebellar development. Chd7 loss leads to a reorientation of the granule cell precursor division plane in the EGL, and it is completely independent of perturbed embryonic granule cell precursor proliferation [99]. Prox1 changes its expression profile with progressive GPC maturation and migration and corresponds to an expression gradient from EGL to IGL that increases from P0 to P5 in mice. The transcription factor is associated with GC differentiation, but it also acts as an identity marker for mature postmitotic cells, as known in the hippocampus [100]. Similar to Chd7, Prox1 was not affected at the transcriptional level by a toxic oxygen insult. Moreover, this concerns Pax2, which is a specific marker for the development of inhibitory cerebellar interneurons [101]. More than three quarters of all inhibitory interneurons of the cerebellar cortex are formed within the first postnatal week, up to P5 [102]. The hypothesis is also that inhibitory interneuronal precursors initiate Pax2, and this appears to be involved in the regulation of GPC-oriented cell division [103,104]. It appears that an impairment of Pax2-expressing interneurons by subsequent hyperoxia at P6 is rather minor.

The synaptic connections between GC and PC during postnatal neurogenesis are generally considered to be an essential site of plasticity [105]. It is understandable that the proliferation capacity and plasticity are related during dynamic cerebellar development [106]. Thus, an existing correlation between the proliferative capacity of the cerebellum and the expression of Syp in the rat was demonstrated in a hypoxia-ischemia model [107]. The cerebellar nuclei cells (CN) and UBCs migrate tangentially beneath the surface of the cerebellum and possibly express various transcription factors, including Tbr1 and Tbr2, which promote migration from the anterior end of the cerebellum to the interior of the cerebellum [108,109,110]. Hyperoxia exposure to the developing cerebellum reduced these two transcription factors close in time to the toxic insult, which was also evident for the pluripotent transcription factor Sox2. Sox2 regulates neural lineages during brain development, as it is important in controlling the proliferation and differentiation of neural progenitor cells [111]. During mouse embryogenesis, Sox2 is expressed in a wide range of differentiated cerebellar cells, such as Bergmann glial cells, and it plays a crucial role in the migration of cerebellar Purkinje cells and granule cells [112]. An adult Sox2 mutant mosaic mouse developed motor defects that were probably based on malformations of the cerebellar granular and molecular cell layers. As demonstrated by Ahlfeld and their colleagues [113], postnatal GC migration is dependent on Sox2, and they suggest that Sox2 is highly associated with the cerebellar network, and that along with BDNF, NT3, and Sema6a, it is responsible for the correct localization of granule cell precursors [75,114].

Premature and newborn infants frequently require sedation for diagnostic and therapeutic procedures, although this may be associated with higher risks if performed under full sedation. The neurotoxic effects of anesthetics also cause cognitive deficits and behavioral problems [115,116]. Given the trade-off between exposures of the preterm brain to potentially neurotoxic anesthetics [117], sedation with a drug that has minimal neurotoxicity, with low apnea risk and cardiovascular stability, may be the possible alternative for high-risk infants [116,118]. Dexmedetomidine displayed high efficacy in pediatric use as a sole sedative for short-term anesthesia, and it appears to have fewer neurotoxic effects compared to the currently used sedatives [119]. The neuroprotective effects of DEX have been demonstrated in both preclinical and clinical studies [43,44,120]. However, there are only a few clinical trials due to the risks for extreme and very preterm infants. The drug is currently used off-label, but has demonstrated advantages over conventional sedatives [121]. DEX demonstrated a shorter treatment time per se, as well as a shorter time of mechanical ventilation, compared to fentanyl in premature infants [39]. Frequently, a reduction in additional midazolam or morphine administration can be detected [40,122], though the frequency of bradycardia in preterm infants has also increased [123,124]. Due to the discussed neuroprotective effect of DEX, an anti-oxidative effect should be primarily discussed. Oxidative stress, associated with elevated ROS levels, which is unavoidable in preterm infants, causes oxidative damage at the cellular and organ level and thus leads to a variety of diseases.

In reference to our data on hyperoxia damage to the developing cerebellum, DEX pretreatment proved to significantly reduce primary cellular impairments. DEX provided protection against oxidative damage in terms of the number of PCs and the shortening of dendrites. The number of PCs and shortened dendrites were protected from oxidative damage by DEX. For the severely damaged mitotic granule cells, this protection by DEX was acutely and sustainably demonstrated under hyperoxia. The proliferation capacity and granule cell-associated factors of cerebellar neurogenesis protect directly after hyperoxia insult and could significantly improve the trailing effects of oxidative stress in terms of the differentiation and migration of GCs. DEX did not affect the factors of mature GCs (Pax2 and Sox2), the neuronal mediators of CN and UBCs (Tbr1 and Tbr2), or the neurotrophic factors under the force of DEX. As just mentioned, a primary anti-oxidative effect of DEX would be understandable, since this effect has already been shown in experimental models. In experimental models of hypoxic brain injury and anesthetic neurotoxicity in neonates, the neuroprotective effects of DEX were found with a reduction in neuronal apoptosis and a decrease in neurological deficits [125,126]. As also shown in previous work, DEX had a neuroprotective effect on hippocampal neurogenesis after oxygen injury [43,44]. Other mechanisms are worth mentioning and should be discussed. Mechanistic studies have clearly demonstrated that DEX serves a variety of signaling pathways. When examined further, DEX is often involved in signaling pathways that are activated under oxidative stress and triggered by various stressors [127]. Additionally, what is worth mentioning are microRNAs (miRNAs), which play an important role in post-transcriptional gene regulation [128]. Many known miRNAs exist in the brain, some of them exclusively in neuronal tissue [129], and in DEX-affected miRNAs with demonstrated neuroprotection [130,131].

Not surprisingly, considering that DEX is a sedative and that it can cross the blood-brain barrier, possible modulatory effects pertain to neuronal processes or neurodegeneration. DEX alone showed an influence on PC dendritogenesis 24 h after application to P7 and reduced the transcription of factors of cerebellar migration, proliferation, differentiation, and maturation. These effects were not sustained. Dexmedetomidine produces sedation by mechanisms that are different from GABAergic anesthetics. It acts selectively on the α2-adrenergic receptors of the locus coeruleus with activation of the inhibitory outputs to the arousal centers [132]. The neurophysiological mechanisms of dexmedetomidine mimic the activation of the brain stem and normal sleep pathways rather than cortical suppression [133]. It is possible that DEX modulates excitatory glutamatergic signaling [134].

Above all, it can be stated that high oxygen concentrations damage the PCs and even more the mitotic precursors of the GC. DEX can counteract this toxic oxygen effect, but under normoxia alone, it also causes short-term impairments of cerebellar neurogenesis. In addition to the antioxidant effect of DEX, it is also important to consider the timing of the damage in relation to brain development. In rodents, the main granule cell population is formed between P1 and P18 in the EGL, whereas the proportion of granule cells in the IGL are formed mainly between P4 and P15 [101]. The passage time of granule cells migrating through the EGL and ML of P10 mice takes up to 48 h, depending on whether initial cell division has already occurred before the onset of migration. At the time of injury of newborn rat pups at P6, the damage by both oxygen and DEX could be transient, affecting the descendant progenitor pool in proliferation and migration in the short term but compensating for it in the long term. What long-term consequences this sensitive, short-term damage might have on plasticity should be investigated in subsequent studies.

## 5. Conclusions

In pediatric anesthesia, the most common anesthetics are those used in adult sedation. With an overall higher anesthetic risk in preterm and neonates, as well as a higher perioperative morbidity and mortality due to immature organ systems, the characterization of sedatives with possible neuroprotective effects is essential. With regard to the safety of studies in preterm infants, the mechanisms of action derived from preclinical models must be considered in light of retrospective human clinical data indicating the potential for long-term neurologic damage. The results of this study suggest that short-term hyperoxia affects the process of cerebellar neurogenesis, but it can be neuroprotected by DEX. The approach used in this study is useful to expand the understanding of cerebellar neurogenesis, and the results may contribute to the elucidation of brain damage caused by prenatal hyperoxia.

## Figures and Tables

**Figure 1 antioxidants-12-00980-f001:**
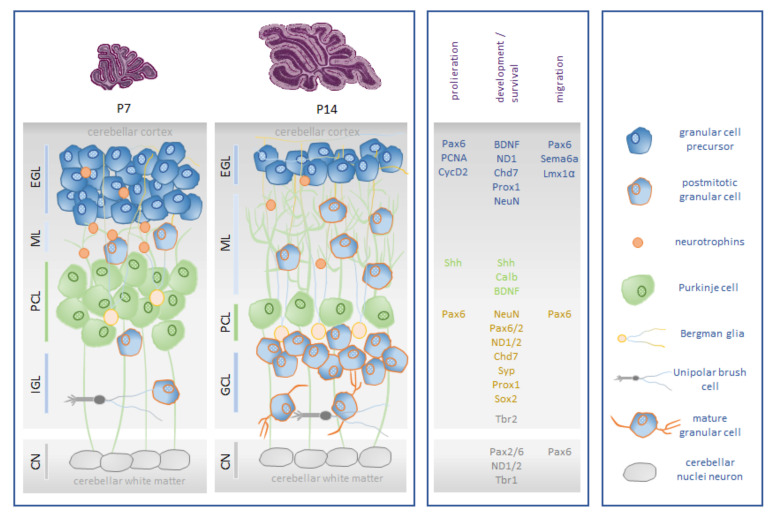
Schematic representation of rodent postnatal granular cell neurogenesis. (**Box left**): The major cell type in the cerebellar cortex is the granule cell (GC). Its precursors, the granular progenitor cells (GPCs), migrate to the external granular layer (EGL), which has high mitotic activity. From E15 until the third postnatal week, GCPs inside the EGL are mitotically active. Onset of differentiation of GCPs into mature GCs initiates around P5. They reach their final destination by radial migration along the Bergmann glia into the internal granular layer (IGL) over the molecular layer (ML) around P20. Unipolar brush cells (UBCs) are an interneuron subtype that are localized in the granule cell layer of the cerebellum. The cerebellar nuclei (CN) are the major initial structures of the cerebellum. In rodents, the proliferation of cerebellar interneurons and granule cells arising from the EGL continues for a period of roughly three weeks [17,27]. (**Box middle**): Cerebellar neurogenesis of granular cells is characterized by the expression of neuronal and proliferative markers and is orchestrated by neuronal transcription factors: brain-derived neurotrophic factor (BDNF), calbindin 1 (Calb1), chromodomain helicase DNA-binding protein 7 (Chd7), cyclin dependent kinase 2 (CycD2), LIM homeobox transcription factor 1 alpha (Lmx1α), neuronal differentiation 1/2 (NeuroD 1/2), neuronal nuclei (NeuN), paired box 2/6 (Pax2/Pax6), proliferating cell nuclear antigen (PCNA), prospero homeobox 1 (Prox1), semaphoring 6a (Sema6a), sonic hedgehog signaling molecule (Shh), sex-determining region Y-box 2 (Sox2), synaptophysin (Syp), and T-box brain transcription factor 1/2 (Tbr1/2). (**Box right**): Cellular players in cerebellar granular cell neurogenesis. Illustration adapted from [28].

**Figure 2 antioxidants-12-00980-f002:**
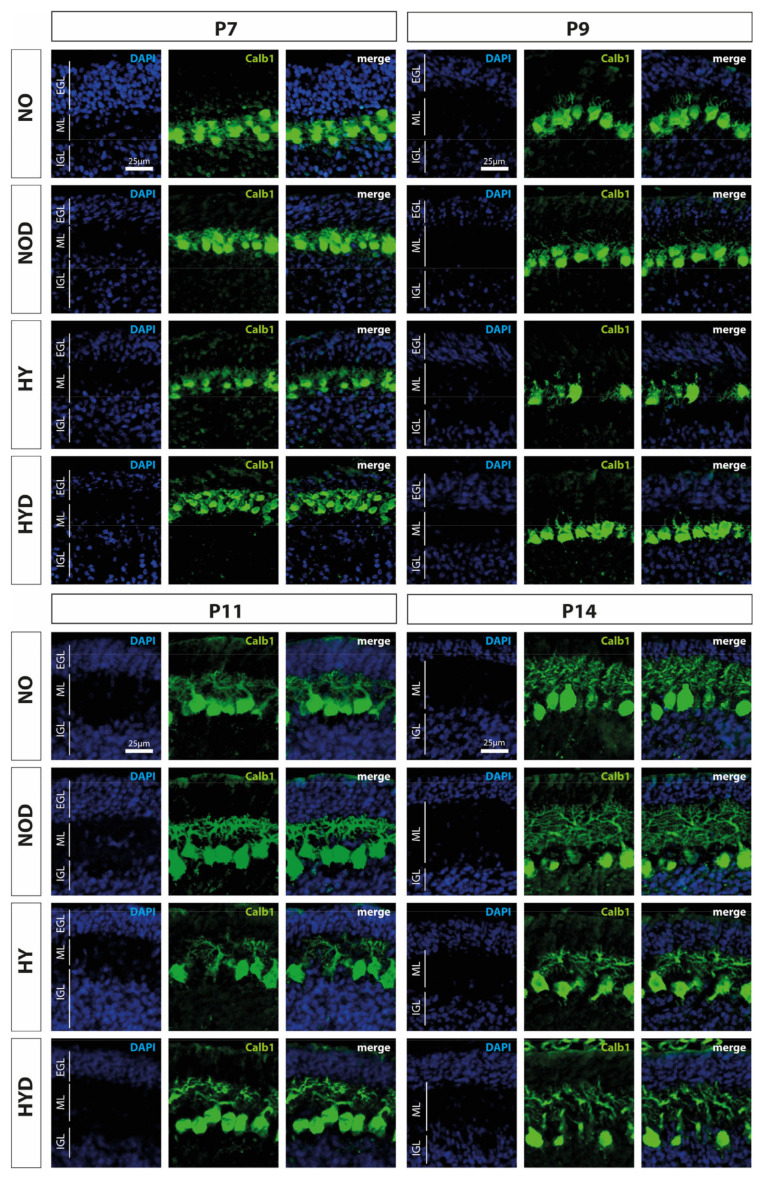
Representative cerebellar paraffin sections co-labeled with calbindin (Calb1, green) and DAPI (blue) of rat pups exposed to normoxia (NO) or hyperoxia (HY) compared to rat pups treated with dexmedetomidine (NOD, HYD). Analyses were conducted for the high oxygen (80%) and normoxia (21%) groups for postnatal day 7 (P7) after 24 h of oxygen exposure and after recovery in room air at P9, P11, and P14. The 24 h of hyperoxia exposure affected the density of Purkinje cells and depth of molecular layer in the newborn rat cerebellum at P7, and persisted until P9 or P11. Dexmedetomidine counteracted these changes. EGL, external granular layer; ML, molecular layer; IGL, internal granular layer. Scale bar 25 μm.

**Figure 3 antioxidants-12-00980-f003:**
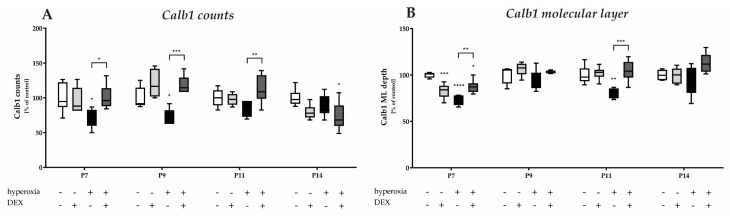
Quantitative analysis of (**A**) calbindin (Calb1) and DAPI-positive Purkinje cells (PC) of the cerebellar molecular layer and (**B**) depth of molecular layer (ML) was performed for the high oxygen (80%) and normal oxygen (21%) groups for 24 h of oxygen exposure from P6 to P7 and after recovery in room air at P9, P11, and P14. The acute hyperoxia exposure (black bars) over one day reduced the number of Purkinje cells and depressed the dendrite depth at P7 and P9 or P11. Dexmedetomidine administration before hyperoxia (dark gray bars) showed a protective effect for calbindin-positive cells (at P7, P9, and P11) and for dendrite length (at P7 and P11). Dexmedetomidine with normoxia exposure (light gray bars) did not affect density of Purkinje cells, but diminished the molecular layer’s depth at P7. Data are normalized to the level of normoxia-exposed rat pups at each time point (control 100%, white bars) and the 100% values are 9.5 (P7), 6.0 (P9), 5.8 (P11), and 5.1 (P14) cells per regions of lobules or 1.5 (P7), 1.4 (P9), 3.1 (P11), and 3.6 (P14) depth of molecular layer, respectively. *n* = 6/group. * *p* < 0.05, ** *p* < 0.01, *** *p* < 0.001, **** *p* < 0.0001 (ANOVA, Bonferroni’s post hoc test).

**Figure 4 antioxidants-12-00980-f004:**
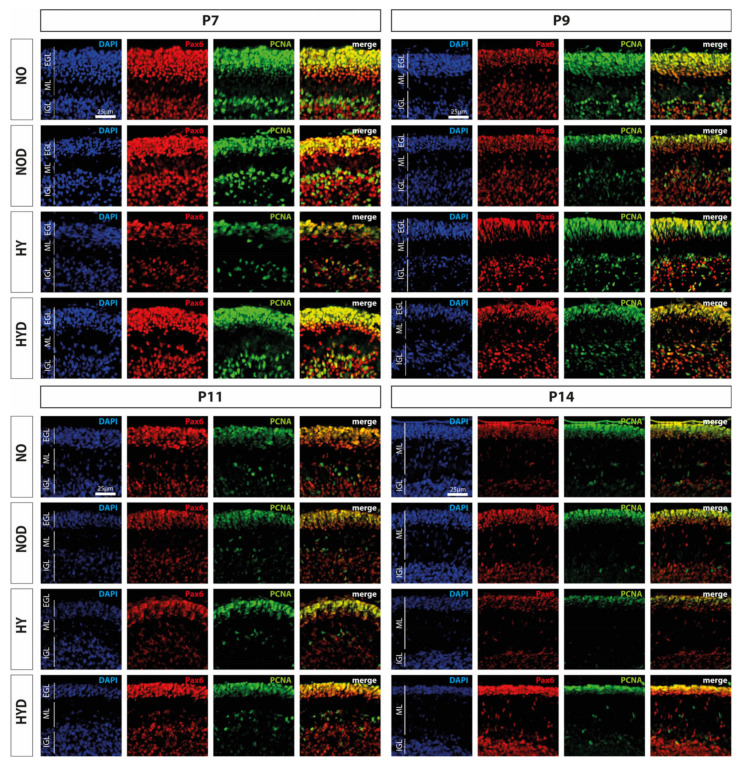
Representative cerebellar paraffin sections co-labeled with Pax6 (red), PCNA (green), and DAPI (blue) of rat pups exposed to normoxia (NO) or hyperoxia (HY) compared to rat pups treated with dexmedetomidine (NOD, HYD). Analyses were conducted for the high oxygen (80%) and normoxia (21%) groups for postnatal day 7 (P7) after 24 h of oxygen exposure and after recovery in room air at P9, P11, and P14. Acute hyperoxia for 1 day reduced granule cell density (Pax6), decreased proliferative capacity (PCNA), and altered EGL thickness in the cerebellum of newborn rats (P7). Dexmedetomidine was able to prevent these changes. EGL, external granular layer; ML, molecular layer; IGL, internal granular layer. Scale bar 25 μm.

**Figure 5 antioxidants-12-00980-f005:**
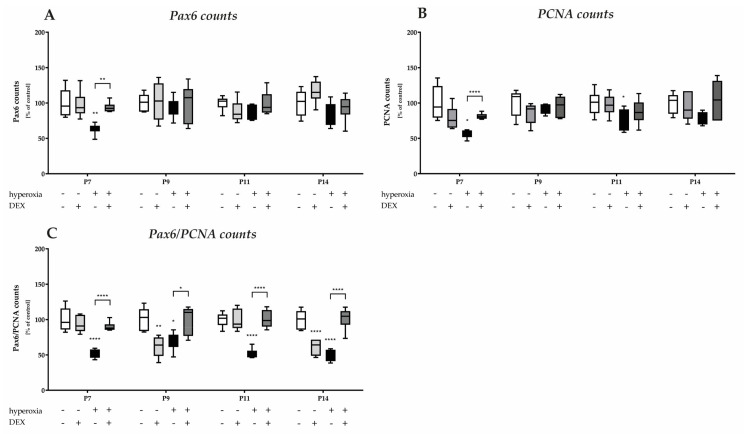
Quantitative analysis of co-labeled (**A**) Pax6+, (**B**) PCNA+, and (**C**) Pax6+/PCNA+ counts of the cerebellar molecular layer was performed for the high oxygen (80%) and normal oxygen (21%) groups for 24 h of oxygen exposure at P7 and after recovery in room air at P9, P11, and P14. The acute hyperoxia exposure (black bars) over one day reduced the number of Pax6+ granular cells (GC) and diminished the proliferating cells (PCNA) of GCs. Pretreatment with dexmedetomidine prior hyperoxia (dark gray bars) was protective for Pax6 positive cells at P7 and for proliferating capacity at P7 and P11. Drastic reductions in mitotic neuronal GC precursors (Pax6+/PCNA+) after hyperoxia persisted after 24 h, immediately after termination, and persisted until P14. Dexmedetomidine was able to reverse this reduced differentiation and maturation after pretreatment at all times. Dexmedetomidine treatment under ambient air (light gray bars) did not affect the density of GCs and proliferation per se, except the mitotic GC precursors at P9 and P14. Data are normalized to the level of normoxia-exposed rat pups at each time point (control 100%, white bars) and the 100% values are 61.1 (P7), 67.6 (P9), 63.0 (P11), and 67.2 (P14) Pax6+ cells, and 41.9 (P7), 47.8 (P9), 47.8 (P11), and 47.9 (P14) PCNA+ cells, or 34.1 (P7), 49.8 (P9), 12.3 (P11), and 13.6 (P14) Pax6+/PCNA+ cells per regions of lobules, respectively. *n* = 6/group. * *p* < 0.05, ** *p* < 0.01, **** *p* < 0.0001 (ANOVA, Bonferroni’s post hoc test; Kruskal-Wallis, Dunn´s post hoc test).

**Figure 6 antioxidants-12-00980-f006:**
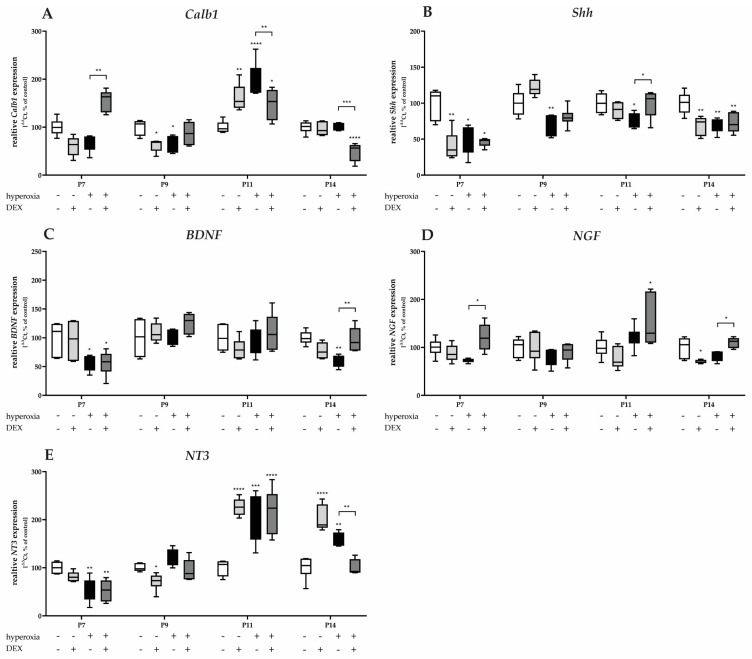
Quantitative analysis of cerebellar homogenates for Purkinje cell-associated factors and neurotrophins of (**A**) *Calb1*, (**B**) *Shh*, (**C**) *BDNF*, (**D**) *NGF*, and (**E**) *NT3* was performed for the high oxygen (80%) and normoxia (21%) groups for 24 h of oxygen exposure at P7 and after recovery in room air at P9, P11, and P14. Analysis of mRNA expressions is depicted for hyperoxia group (black bars), hyperoxia with prior dexmedetomidine administration (deep gray bars), and dexmedetomidine under ambient air exposure (light gray bars) in comparison to normoxia vehicle-treated control animals (white bars). Data are normalized to the level of normoxia-exposed rat pups at each time point (control 100%, white bars). *n* = 6/group. * *p* < 0.05, ** *p* < 0.01, *** *p* < 0.001, **** *p* < 0.0001 (ANOVA, Bonferroni’s post hoc test; Kruskal-Wallis, Dunn’s post hoc test; Brown-Forsythe, Dunnett’s post hoc test).

**Figure 7 antioxidants-12-00980-f007:**
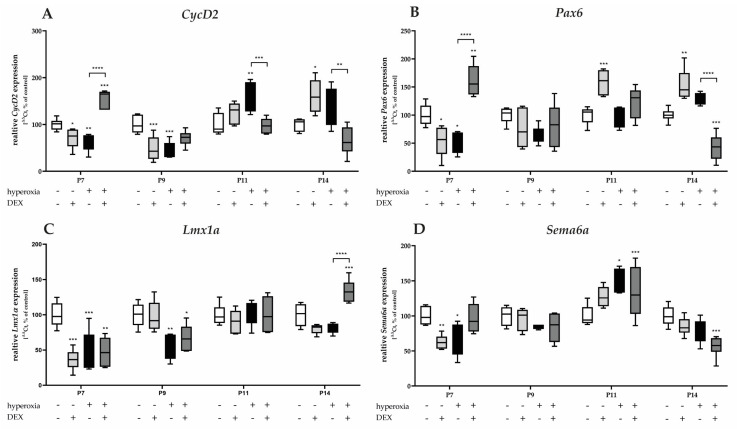
Quantitative analysis of cerebellar homogenates for proliferating and migrating GPC-associated factors of (**A**) *CycD2*, (**B**) *Pax6*, (**C**) *Lmx1α*, and (**D**) *Sema6a* was performed for the high oxygen (80%) and normoxia (21%) groups for 24 h of oxygen exposure at P7 and after recovery in room air at P9, P11, and P14. Analysis of mRNA expressions is depicted for hyperoxia group (black bars), hyperoxia with prior dexmedetomidine administration (deep gray bars), and dexmedetomidine under ambient air exposure (light gray bars) in comparison to normoxia vehicle-treated control animals (white bars). Data are normalized to the level of normoxia-exposed rat pups at each time point (control 100%, white bars). *n* = 6/group. * *p* < 0.05, ** *p* < 0.01, *** *p* < 0.001, **** *p* < 0.0001 (ANOVA, Bonferroni’s post hoc test; Kruskal-Wallis, Dunn’s post hoc test; Brown-Forsythe, Dunnett’s post hoc test).

**Figure 8 antioxidants-12-00980-f008:**
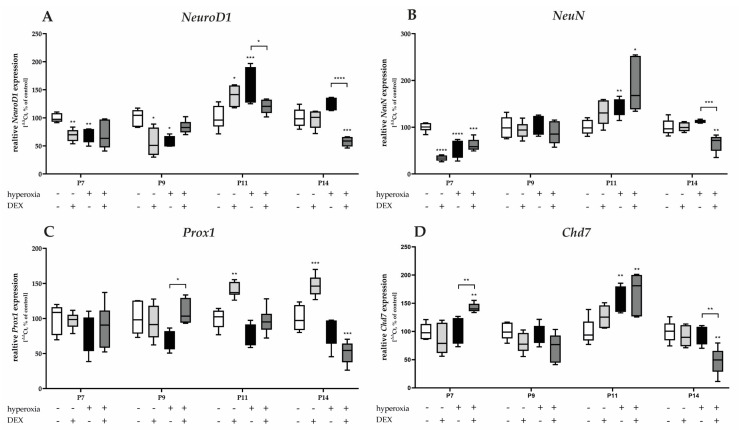
Quantitative analysis of cerebellar homogenates for the survival and development of GPC-associated factors of (**A**) *NeuroD1*, (**B**) *NeuN*, (**C**) *Prox1*, and (**D**) *Chd7* was performed for the high oxygen (80%) and normoxia (21%) groups for 24 h of oxygen exposure at P7 and after recovery in room air at P9, P11, and P14. Analysis of mRNA expressions is depicted for hyperoxia group (black bars), hyperoxia with prior dexmedetomidine administration (deep gray bars), and dexmedetomidine under ambient air exposure (light gray bars) in comparison to normoxia vehicle-treated control animals (white bars). Data are normalized to the level of normoxia-exposed rat pups at each time point (control 100%, white bars). *n* = 6/group. * *p* < 0.05, ** *p* < 0.01, *** *p* < 0.001, **** *p* < 0.0001 (ANOVA, Bonferroni’s post hoc test; Kruskal-Wallis, Dunn’s post hoc test; Brown-Forsythe, Dunnett’s post hoc test).

**Figure 9 antioxidants-12-00980-f009:**
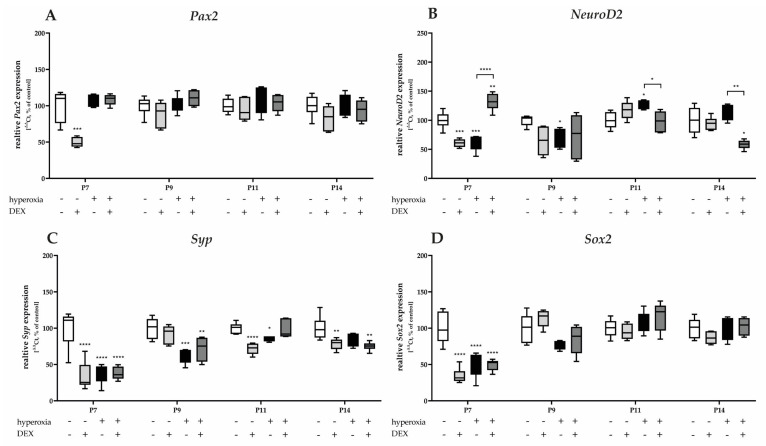
Quantitative analysis of cerebellar homogenates for the survival and development of GC-associated mediators of (**A**) *Pax2*, (**B**) *NeuroD2*, (**C**) *Syp*, and (**D**) *Sox2* was performed for the high oxygen (80%) and normoxia (21%) groups for 24 h of oxygen exposure at P7 and after recovery in room air at P9, P11, and P14. Analysis of mRNA expressions is depicted for hyperoxia group (black bars), hyperoxia with prior dexmedetomidine administration (deep gray bars), and dexmedetomidine under ambient air exposure (light gray bars) in comparison to normoxia vehicle-treated control animals (white bars). Data are normalized to the level of normoxia-exposed rat pups at each time point (control 100%, white bars). *n* = 6/group. * *p* < 0.05, ** *p* < 0.01, *** *p* < 0.001, **** *p* < 0.0001 (ANOVA, Bonferroni’s post hoc test; Kruskal-Wallis, Dunn’s post hoc test; Brown-Forsythe, Dunnett’s post hoc test).

**Figure 10 antioxidants-12-00980-f010:**
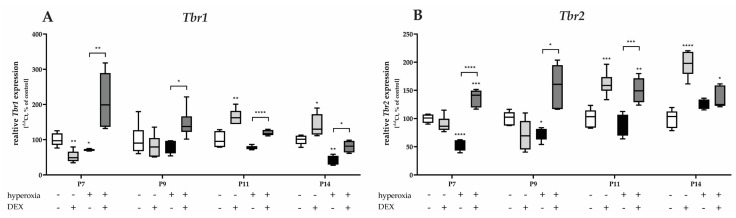
Quantitative analysis of cerebellar homogenates for UBC- and CN-associated factors of (**A**) *Tbr2* and (**B**) *Tbr1* was performed for the high oxygen (80%) and normoxia (21%) groups for 24 h of oxygen exposure at P7 and after recovery in room air at P9, P11, and P14. Analysis of mRNA expressions is depicted for hyperoxia group (black bars), hyperoxia with prior dexmedetomidine administration (deep gray bars), and dexmedetomidine under ambient air exposure (light gray bars) in comparison to normoxia vehicle-treated control animals (white bars). Data are normalized to the level of normoxia-exposed rat pups at each time point (control 100%, white bars). *n* = 6/group. * *p* < 0.05, ** *p* < 0.01, *** *p* < 0.001, **** *p* < 0.0001 (ANOVA, Bonferroni’s post hoc test; Kruskal-Wallis, Dunn’s post hoc test; Brown-Forsythe, Dunnett’s post hoc test).

**Table 1 antioxidants-12-00980-t001:** Sequences of oligonucleotides.

	Oligonucleotide Sequence 5′−3′	Accession No.
BDNF		
forward	TCAGCAGTCAAGTGCCTTTGG	NM_012513.4
reverse	CGCCGAACCCTCATAGACATG	
probe	CCTCCTCTGCTCTTTCTGCTGGAGGAATACAA	
Calb1		
forward	CGACGCTGATGGAAGTGGTT	NM_031984.2
reverse	TCCAATCCAGCCTTCTTTCG	
probe	AAGGAAAGGAGCTGCAGAA	
Chd7		
forward	CAAGCTTCTGGAGGGACTGAA	NM_001107906.2
reverse	AAGAGCTCCTCCACAGTGTTCTG	
probe	TGGAACACAAAGTGCTGC	
CycD2		
forward	CGTACATGCGCAGGATGGT	NM_199501.1
reverse	AATTCATGGCCAGAGGAAAGAC	
probe	TGGATGCTAGAGGTCTGTGA	
HPRT		
forward	GGAAAGAACGTCTTGATTGTTGAA	NM_012583.2
reverse	CCAACACTTCGAGAGGTCCTTTT	
probe	CTTTCCTTGGTCAAGCAGTACAGCCCC	
Lmx1α		
forward	ACCACTCAGCAGAGGAGAGCAT	NM_001105967.2
reverse	TGTCTCCGCAGCCAGAGTCT	
probe	AAGTATCCTCCAAGCCCT	
NeuroD1		
forward	TCAGCATCAATGGCAACTTC	NM_019218.2
reverse	AAGATTGATCCGTGGCTTTG	
probe	TTACCATGCACTACCCTGCA	
NeuroD2		
forward	TCTGGTGTCCTACGTGCAGA	NM_019326.1
reverse	CCTGCTCCGTGAGGAAGTTA	
probe	TGCCTGCAGCTGAACTCTC	
NeuN		
forward	GCTGAATGGGACGATCGTAGAG	NM_001134498.2
reverse	CATATGGGTTCCCAGGCTTCT	
probe	AGGTCAATAATGCCACGGC	
NGF		
forward	ACCCAAGCTCACCTCAGTGTCT	NM_001277055.1
reverse	GACATTACGCTATGCACCTCAGAGT	
probe	CAATAAAGGCTTTGCCAAGG	
NT3		
forward	AGAACATCACCACGGAGGAAA	NM_031073.3
reverse	GGTCACCCACAGGCTCTCA	
probe	AGAGCATAAGAGTCACCGAG	
Pax2		
forward	GTACTACGAGACTGGCAGCATCAA	NM_001106361.1
reverse	TCGGGTTCTGTCGCTTGTATT	
probe	CCAAAGTGGTGGACAAGA	
Pax6		
forward	TCCCTATCAGCAGCAGTTTCAGT	NM_013001.2
reverse	GTCTGTGCGGCCCAACAT	
probe	CTCCTCCTTTACATCGGGTT	
Prox1		
forward	TGCCTTTTCCAGGAGCAACTAT	NM_001107201.1
reverse	CCGCTGGCTTGGAAACTG	
probe	ACATGAACAAAAACGGTGGC	
Sema6a		
forward	GCCATTGATGCGGTCATTTA	NM_001108430.2
reverse	GTACGGCTCTTTCAACCACTTTG	
probe	CTTGGAGACAGTCCTACC	
Shh		
forward	GCTGATGACTCAGAGGTGCAAA	NM_017221.1
reverse	CCTCAGTCACTCGAAGCTTCACT	
probe	CAAGTTAAATGCCTTGGCCA	
Sox2		
forward	ACAGATGCAGCCGATGCA	NM_001109181.1
reverse	GGTGCCCTGCTGCGAGTA	
probe	CAGTACAACTCCATGACCAG	
Syp		
forward	TTCAGGCTGCACCAAGTGTA	NM_003179.3
reverse	TTCAGCCGACGAGGAGTAGT	
probe	AGGGGGCACTACCAAGATCT	
Tbr1		
forward	TCCCAATCACTGGAGGTTTCA	NM_001191070.1
reverse	GGATGCATATAGACCCGGTTTC	
probe	AAATGGGTTCCTTGTGGCAA	
Tbr2		
forward	ACGCAGATGATAGTGTTGCAGTCT	XM_006226608.2
reverse	ATTCAAGTCCTCCACACCATCCT	
probe	CACAAATACCAACCTCGACT	

Abbreviations: brain-derived neurotrophic factor (BDNF), calbindin 1 (Calb1), chromodomain helicase DNA-binding protein 7 (Chd7), cyclin D2 (CycD2), hypoxanthine-guanine phosphoribosyl-transferase (HPRT), LIM homeobox transcription factor 1 alpha (Lmx1α), neurogenic differentiation 1 (NeuroD1), neurogenic differentiation 2 (NeuroD2), neuronal nuclei (NeuN), nerve growth factor (NGF), neurotrophin 3 (NT3), paired box 2 (Pax2), paired box 6 (Pax6), prospero homeobox 1 (Prox1), semaphoring 6a (Sema6a), sonic hedgehog signaling molecule (Shh), SRY-box transcription factor 2 (Sox2), synaptophysin (Syp), T-box brain transcription factor 1 (Tbr1), and T-box brain transcription factor 2 (Tbr2).

## Data Availability

The data used to support the findings of this study are available from the corresponding author upon request. The analyzed data used to create the graphs and statistical evaluation are attached in the Appendix A of this work.

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
