# Peer review of "Protective Effect of Dexmedetomidine against Hyperoxia-Damaged Cerebellar Neurodevelopment in the Juvenile Rat"

_antioxidants, 2023, doi:10.3390/antiox12040980_

Round 1

Reviewer 1 Report

The authors present a very interesting manuscript convincingly describing the effects of dexmedetomidine on cerebellar cells after hyperoxic insult in juvenile murine.

Hyperoxia is known to produce cellular rections at different levels (Balestra C, Arya AK, Leveque C, Virgili F, Germonpre P, Lambrechts K, Lafere P & Thom SR. (2022). Varying Oxygen Partial Pressure Elicits Blood-Borne Microparticles Expressing Different Cell-Specific Proteins-Toward a Targeted Use of Oxygen? Int J Mol Sci 23, 7888.).

The manuscript is well presented and the rationale adequately documented.

In the manuscript has some typos that can easily be corrected, e.g. : 2.3. Tissue preparation on the fourth line has to be retyped since an inversion is present.

Statistical approach is sound and correct nevertheless, in some figures the statistical tests declared are not the same that have been described in the methods, just add them as well. 

I find the graphs very small, I wonder if some changes could be made to enlarge them.

I totally endorse the discussion and appreciate the neurotoxic description. However there are some works showing that intermittency of oxygen supply may be beneficial and develop, among others, erythropoietin that is also antiapoptotic and neuroprotective (together with neuropoietin). It could be of interest to further investigate some oxygen variations without falling nto hypoxia and see if this could trigger some positive reactions. This is of course just a reflexion based on previous investigations (Balestra C & Kot J. (2021). Oxygen: A Stimulus, Not "Only" a Drug. Medicina (Kaunas) 57.Valacchi G, Virgili F, Cervellati C & Pecorelli A. (2018). OxInflammation: From Subclinical Condition to Pathological Biomarker. Front Physiol 9, 858.)

I do not have more suggestions and would like to thank-you for giving me the opportunity of reviewing such an interesting manuscript.

Reviewer 2 Report

The manuscript from Puls et al. presents the results of pre-treatment with Dexmedetomidine of juvenile rats under hyperoxia as a causative condition for damaged cerebral neurodevelopment. Among their results it is worth to mention:

  1. The authors have demonstrated that Dexmedetomidine was able to reduce the negative effects of hyperoxia on the number of Calbindin1 positive Purkinje cells and their dendrite’s growth.
  2. On the other hand, pre-treatment with Dexmedetomidine prevented de hypoxia reduction of granule cell precursors.
  3. However, Dexmedetomidine has shown only a low effect on hyperoxia-injured cerebral development mediators, after analysis of the transcription expression of NeuN, NeuroD1 and Chd7. 

In summary, the experiments carried out by Puls at al. provide consistent evidence that Dexmedetomidine has neuroprotectant activity in those physio-pathological situations of the neurodevelopment were hypoxia is a disrupting condition.

This manuscript deserves to be accepted for publication in ANTOXIDANTS.

Reviewer 3 Report

The article titled "Protective Effect of Dexmedetomidine against Hyperoxia-Damaged Cerebral Neurodevelopment in the Juvenile Rat" is a well-designed study investigating the neuroprotective effects of dexmedetomidine in juvenile rats exposed to hyperoxia. The study provides important insights into the molecular mechanisms underlying the effects of hyperoxia on cerebellar development and offers promising therapeutic potential for this critical condition.
The authors present their methodology clearly and concisely, allowing easy interpretation of the results. The experimental design is thorough and comprehensive, with appropriate control groups and statistical analyzes. The results are convincing and supported by sufficient evidence. The manuscript is well written and organized, making it easy to follow the experimental results and conclusions.
Although the article is well presented, there are some minor comments that could help improve the perception of the article. The reviewer feels that the presentation of the cell quantification data as a percentage of the control group is unfortunate and does not allow for a critical evaluation of cellularity. It is suggested that the data be presented as absolute number of cells per unit area.
In the "Materials and Methods" section, you must report the number of pups in each group. The measurement of the length of the dendrites of the Purkenje cells needs to be described in more detail. Indicate which is shown in Figure 3b, the length of the processes of the Purkenje cells or the depth of the molecular layer?
The authors measured gene expression in cerebellar tissue, from which they draw conclusions about the expression of certain genes, e.g., BDNF, in Purkinje cells?
